# Signature of type-II Weyl semimetal phase in MoTe$_2$

J. Jiang[1,2,3,4,*], Z.K. Liu[1,*], Y. Sun[5,*], H.F. Yang[2,6,*], C.R. Rajamathi[5], Y.P. Qi[5], L.X. Yang[7], C. Chen[2], H. Peng[2], C.-C. Hwang[4], S.Z. Sun[8], S.-K. Mo[3], I. Vobornik[9], J. Fujii[9], S.S.P. Parkin[10], C. Felser[5], B.H. Yan[5] & Y.L. Chen[1,2,7,8]

Topological Weyl semimetal (TWS), a new state of quantum matter, has sparked enormous research interest recently. Possessing unique Weyl fermions in the bulk and Fermi arcs on the surface, TWSs offer a rare platform for realizing many exotic physical phenomena. TWSs can be classified into type-I that respect Lorentz symmetry and type-II that do not. Here, we directly visualize the electronic structure of MoTe$_2$, a recently proposed type-II TWS. Using angle-resolved photoemission spectroscopy (ARPES), we unravel the unique surface Fermi arcs, in good agreement with our *ab initio* calculations that have nontrivial topological nature. Our work not only leads to new understandings of the unusual properties discovered in this family of compounds, but also allows for the further exploration of exotic properties and practical applications of type-II TWSs, as well as the interplay between superconductivity (MoTe$_2$ was discovered to be superconducting recently) and their topological order.

[1] School of Physical Science and Technology, ShanghaiTech University and CAS-Shanghai Science Research Center, Shanghai 201203, People's Republic of China. [2] Department of Physics, University of Oxford, Oxford OX1 3PU, UK. [3] Advanced Light Source, Lawrence Berkeley National Laboratory, Berkeley, California 94720, USA. [4] Pohang Accelerator Laboratory, POSTECH, Pohang 790-784, Korea. [5] Max Planck Institute for Chemical Physics of Solids, D-01187 Dresden, Germany. [6] State Key Laboratory of Functional Materials for Informatics, SIMIT, Chinese Academy of Sciences, Shanghai 200050, People's Republic of China. [7] State Key Laboratory of Low Dimensional Quantum Physics, Department of Physics and Collaborative Innovation Center of Quantum Matter, Tsinghua University, Beijing 100084, People's Republic of China. [8] Hefei Science Center, CAS and SCGY, University of Science and Technology of China, Hefei 200026, People's Republic of China. [9] Istituto Officina dei Materiali (IOM)-CNR, Laboratorio TASC, Trieste 34149, Italy. [10] Max Planck Institute of Microstructure Physics, Halle D-06120, Germany. * These authors contributed equally to this work. Correspondence and requests for materials should be addressed to Y.L.C. (email: yulin.chen@physics.ox.ac.uk).

Three-dimensional (3D) topological Weyl semimetals (TWSs) are novel topological quantum materials discovered and intensively investigated recently because of their intimate link between concepts of different fields of physics and material science, as well as the broad application potential[1–6]. In a TWS, low-energy electronic excitations form composite Weyl fermions dispersing linearly along all the three momentum directions across the Weyl points (WPs)[7–10] that always appear in pairs with opposite chirality. Between WPs of different chirality, there exist intriguing surface Fermi arcs, the unconventional open curve-like Fermi surfaces (FSs) with nondegenerate spin texture. The exotic bulk and surface electronic structures thus provide an ideal platform for many novel physical phenomena, such as negative magnetoresistance, anomalous quantum Hall effect and chiral magnetic effects[11–20].

Interestingly, the TWSs can be further classified into two types by Fermiology and whether the Lorentz symmetry is respected: type-I TWS that hosts point-like bulk FSs formed solely by WPs that approximately respect the Lorentz symmetry[4–6]; and type-II TWS that breaks the Lorentz symmetry and harbours finite electron density of states at the Fermi energy . Recently, type-I TWSs have been discovered in the (Ta, Nb) (As, P) family of compounds[7–9,21,22]. However, the complex 3D crystal structure in these first-generation TWSs may pose difficulties in the exploration of their practical applications, and their large number (12 pairs) of WPs could make the exploration of novel physical phenomena complicated. Therefore, new TWS materials easier to process, more friendly to device fabrication applications and with less WPs are desired.

More recently, type-II TWS has been proposed to exist in layered transition metal dichalcogenides (TMDs, for example, $WTe_2$, $MoTe_2$ and $W_xMo_{1-x}Te_2$)[23–26]. The distribution of the WPs is found to be very sensitive to the lattice parameters. Even a small difference of the lattice constant ($\sim 0.5\%$) will change the number of WPs in the Brillouin zone (BZ) and lead to different topology of the surface Fermi arcs[23,27]. With less WPs and nonvanishing electronic density of states at the Fermi surface, type-II TWSs in TMD compounds are expected to show very different properties from the type-I TWSs, such as anisotropic chiral anomaly depending on the current directions, a novel anomalous Hall effect[24]. In addition, the layered nature of the TMD TWSs greatly facilitates the fabrication of devices[28], making them an ideal platform for the realization of novel TWS applications.

In this work, using high-resolution angle-resolved photoemission spectroscopy (ARPES), we systematically study the electronic structure of the orthorhombic ($T_d$)-$MoTe_2$. By carrying out broad-range photon energy-dependent measurements, we can conclude that the observed surface Fermi arcs are consistent with previous[23] and our ab initio calculations that are topologically nontrivial. The discovery of the TWS phase in $MoTe_2$ could also help understand the puzzling physical properties in the orthorhombic phase of TMDs recently observed[29], and provide a more feasible material platform for the future applications of TWSs because of their layered structures. Furthermore, with the recent discovery of superconductivity in $MoTe_2$ (ref. 25), it even provides an ideal platform for the study of interplays between superconductivity and the nontrivial topological order.

## Results

**Sample characterization.** The illustration of the type-I and type-II Weyl fermions is shown in Fig. 1a. Figure 1b,c shows the distributions of the WPs of $T_d$-$MoTe_2$ in the 3D momentum pace and momentum–energy space, respectively. Figure 1d,e present the 3D BZ of $T_d$-$MoTe_2$ and its projected surface BZ with calculated Fermi surface. The crystal structure of $T_d$-$MoTe_2$ is illustrated in Fig. 1f, clearly showing an alternating stacking of Te-Mo-Te triple layer structure (space group $P_{mn21}$). We have synthesized high-quality single crystals for this work: the large flat shinning surface (Fig. 1g, left inset) after in situ cleaving is ideal for ARPES measurements; and the crystal symmetry is verified by the Laue (before cleaving) and low-energy electron diffraction (after cleaving) measurements (Fig. 1g, right inset). The lattice constants of the single crystal here are verified by the X-ray diffraction method to be $a = 6.335$ Å, $b = 3.477$ Å, $c = 13.883$ Å that will result in 8 WPs in the $k_z = 0$ plane according to ab inito calculation[23,25]. The core-level photoemission spectrum clearly shows the characteristic $Te_{4d}$ and $Mo_{4p/4s}$ peaks (Fig. 1g), and the broad Fermi surface mapping in Fig. 1h covering multiple BZs confirmed the (001) cleave plane with correct lattice constant. In Fig. 1h, one can already see that the shape of the FS is in general agreements with our calculation (Fig. 1e).

**General electronic structure of $T_d$-$MoTe_2$.** To investigate the detailed electronic structures, we focus on one BZ by carrying out high-resolution ARPES measurements, and the results are demonstrated in Fig. 2. In Fig. 2a, the semimetallic nature of $T_d$-$MoTe_2$ can be clearly seen, where large hole pockets are centred around $\bar{\Gamma}$ and electron pockets are away from $\bar{\Gamma}$ along the $\bar{\Gamma}\bar{Y}$ direction. The evolution of these pockets with binding energy is illustrated in the dispersion plots (Fig. 2b–e) and the bands' constant energy contours (Fig. 2f–o). In Fig. 2b,c, the strongly anisotropic band dispersion along the $\bar{\Gamma}\bar{X}$ and $\bar{\Gamma}\bar{Y}$ directions are clearly shown: the hole- and electron-like Fermi crossings are both observed along $\bar{\Gamma}\bar{Y}$ direction, whereas only hole-like Fermi crossings can be observed along $\bar{\Gamma}\bar{X}$ that show broad agreement with our ab initio calculations (Fig. 2d,e). The electron- and hole-pocket evolution can be better seen in Fig. 2f–j, where the outer electron pockets shrink with $E_B$ and disappear beyond 0.075 eV, consistent with the dispersion in Fig. 2b,c and also our ab initio calculations (see Fig. 2k,o, and more comparisons between our experimental result and calculations can be found in Supplementary Note 1 and Supplementary Figs 1–3). The evolution of the constant energy contours through a larger energy scale is illustrated in Fig. 2p, showing a rich evolution of texture from multiple bands.

**Surface and bulk electronic structures of $T_d$-$MoTe_2$.** In order to distinguish the surface and bulk contributions from numerous band dispersions in Fig. 2, we conducted photon energy-dependent ARPES experiment[30] with a broad photon energy range of 20–120 eV, covering more than 5 BZs along the $k_z$ direction (see Fig. 3a, and more details can be found in Supplementary Note 2 and Supplementary Fig. 4). In Fig. 3a, throughout the whole range, bands with negligible $k_z$ dispersion can be observed, showing a clear sign of surface states. By plotting the FSs measured with different photon energies (Fig. 3b–i), one can clearly see that the nondispersive bands along $k_z$ in Fig. 3a correspond to the FS features in the narrow region between the bulk and electron pockets (as highlighted by the red rectangles in Fig. 3b–i) that show identical shape under different photon energies, whereas the pockets from the bulk states vary strongly (as also can be seen by the dispersive bands in Fig. 3a). This observation is again in good agreement with the ab initio calculation, as shown in Fig. 3j, where the sharp surface state bands clearly emerge between the electron and hole pockets.

**Surface Fermi arcs in $T_d$-$MoTe_2$.** After establishing the overall band structures and the identification of surface state regions, we further zoom in to these regions to search for the characteristic surface Fermi arcs, a hallmark of the electronic structure of a

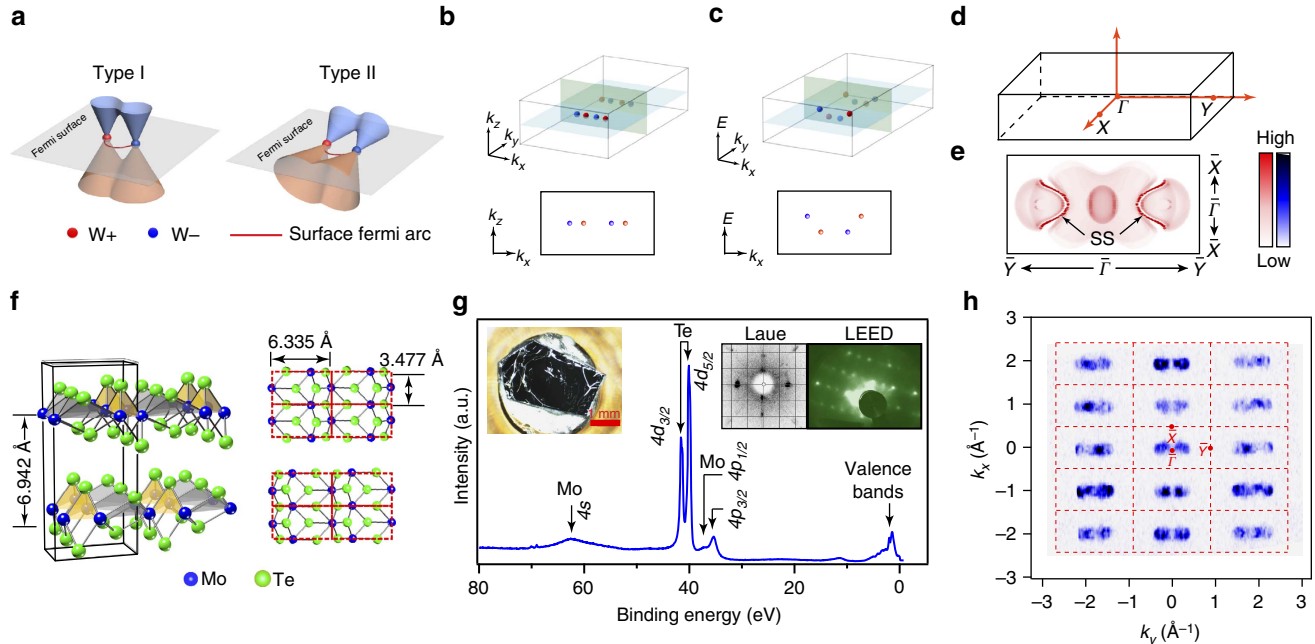

**Figure 1 | Basic characteristics of type-II TWS and characterization of MoTe₂ single crystals.** (**a**) Schematic illustration of type-I and type-II Weyl fermions and Weyl points in the momentum space. (**b**) Schematic showing the Weyl points of MoTe₂ in a BZ and its projection to the $k_x$–$k_z$ plane. (**c**) Schematic showing the Weyl points in the energy–momentum space and their projection to the $k_x$–$E$ plane. (**d**) Illustration of the BZ of $T_d$-MoTe₂ with high symmetry point marked. (**e**) Fermi surface by our *ab initio* calculation, showing the bulk electron, hole pockets and the surface states (marked as SS). (**f**) Crystal structure of the $T_d$-MoTe₂, showing alternating '…A-B-A-B…' stacking of MoTe₂ layers. (**g**) Core-level spectrum showing characteristic Mo₄ₛ, Te₄ₚ/₄d core-level peaks. Left inset: cleaved surface of MoTe₂ single crystal used in this study. Right insets: Laue and low-energy electron diffraction (LEED) pattern showing the $T_d$ phase of MoTe₂. (**h**) Broad-range photoemission spectral intensity map of the Fermi surface (FS) covers >10 BZs, showing the correct symmetry and the characteristic FS.

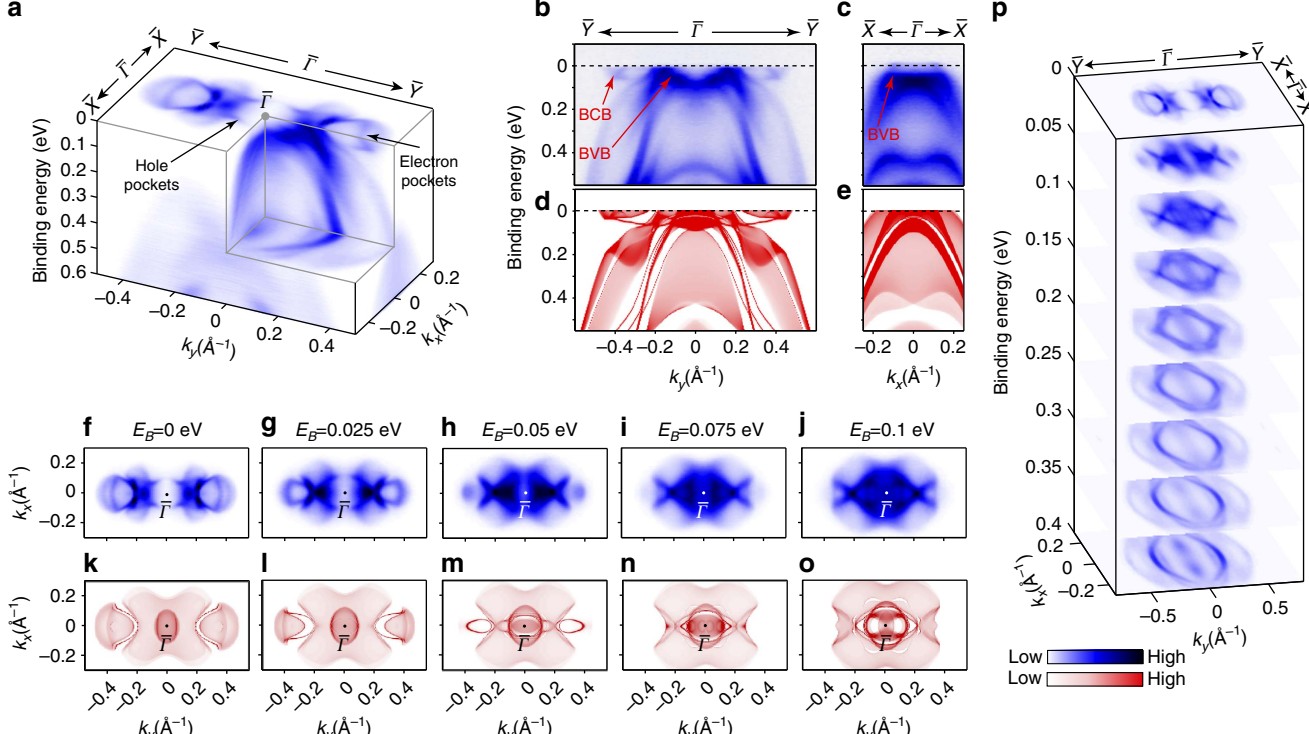

**Figure 2 | General electronic structure of $T_d$-MoTe₂.** (**a**) The 3D intensity plot of the photoemission spectra around $\bar{\Gamma}$, with the electron and hole pockets indicated. (**b,c**) High symmetry cut along the $\bar{Y}\bar{\Gamma}\bar{Y}$ and $\bar{X}\bar{\Gamma}\bar{X}$ directions, respectively. BCB, bulk conduction band; BVB, bulk valance band. (**d,e**) The corresponding calculations along the $\bar{Y}\bar{\Gamma}\bar{Y}$ and $\bar{X}\bar{\Gamma}\bar{X}$ directions, respectively. (**f–j**) Photoemission spectral intensity map showing the constant energy contours of bands at $E_B = 0$, 0.025, 0.05, 0.075 and 0.1 eV, respectively. (**k–o**) Corresponding calculated constant energy contours at the same binding energies as in experiments above. (**p**) Stacking plots of constant-energy contours in broader binding energy range show the band structure evolution.

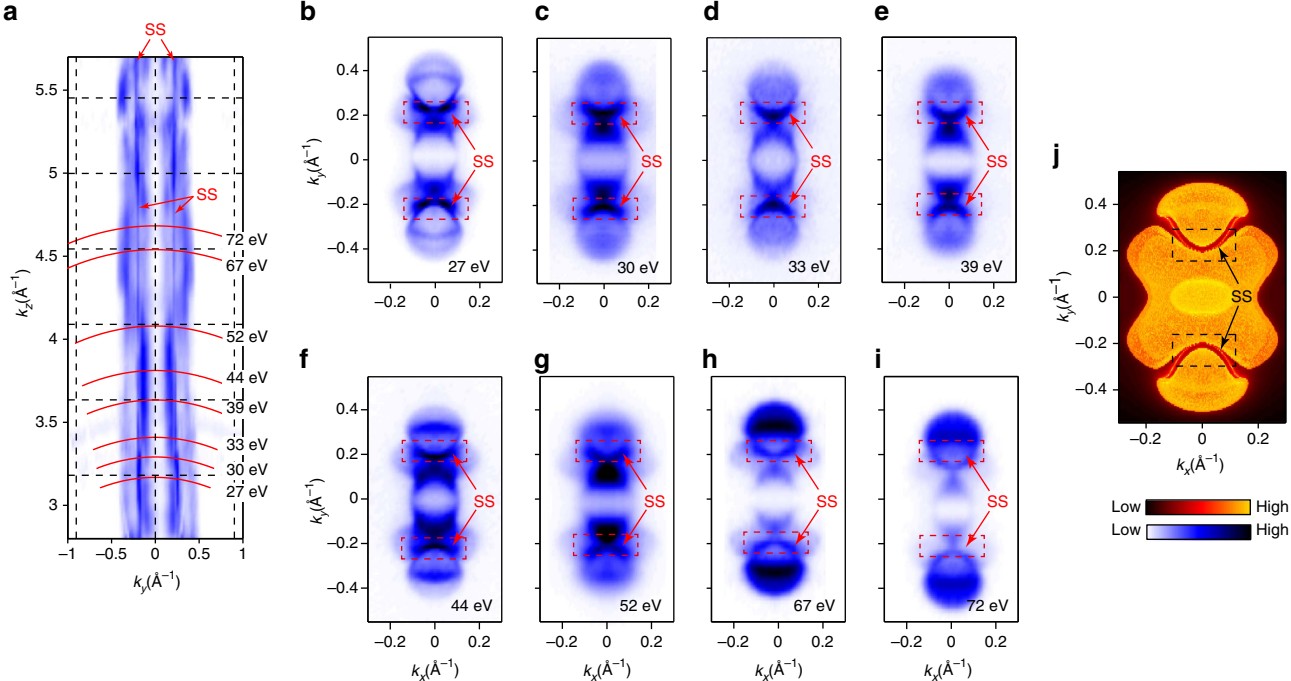

**Figure 3 | Bulk and surface electronic states of MoTe₂ probed by photon energy-dependent ARPES measurements. (a)** The photoemission spectral intensity map in $k_y$–$k_z$ plane, where the red arrows indicate the surface states (marked as SS) that show no dispersion along the $k_z$ direction. Red curves indicate the $k_z$ locations of different photon energies. **(b–i)** Fermi surfaces measured under different photon energies as indicated in **a**. Red dashed rectangles and arrows highlight the position of the surface states that do not change shape with the photon energies. **(j)** Calculated Fermi surface with black dashed rectangles highlighting the same area as in **b-i** indicating clear surface states inside.

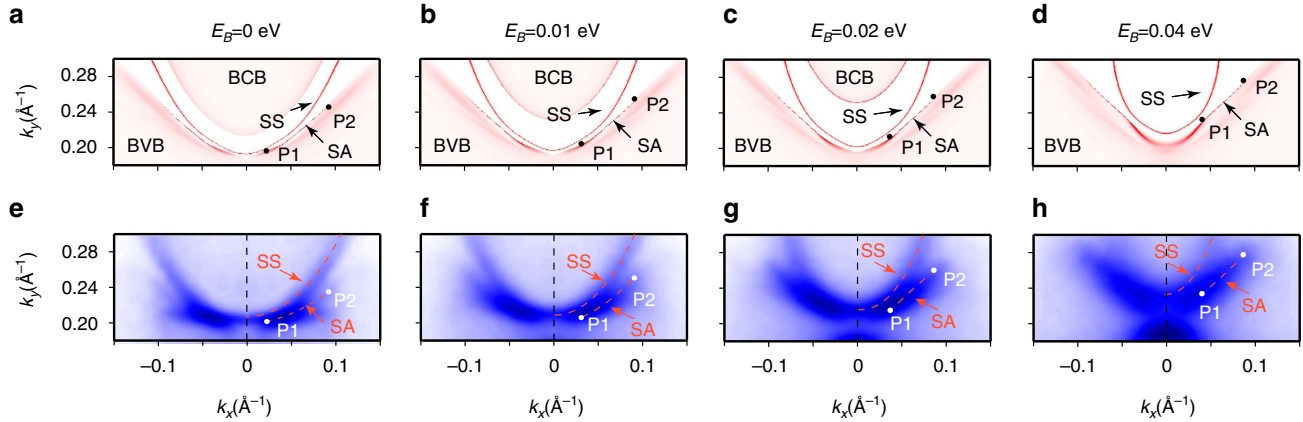

**Figure 4 | Evolution of the Fermi arcs with binding energies. (a–d)** Evolution of the projection of the Fermi arcs in the calculation at $E_B = 0$, 0.01, 0.02 and 0.04 eV, respectively. The regular surface state and the surface Fermi arc are labelled as SS and SA, respectively; points P1 and P2 indicate the starting and ending points of the surface Fermi arcs. **(e–h)** Photoemission intensity map in the same area as in **a–d**. The dashed lines show the surface state (SS) and the surface Fermi arcs (SA), corresponding to those in **a–d**.

TWS. According to the *ab initio* calculations, although the WPs lie slightly above the Fermi level (∼26 and 79 meV above the experimental Fermi-level according to the calculations, see ref. 23 and Supplementary Fig. 2 for details), the surface Fermi arcs are still visible, lying within the narrow region between the electron and hole pockets (see Fig. 4a, marked as SA between the points P1 and P2). As there is another surface state (marked as SS) nearby, to best visualize and separate the two surface states, we chose a low (27 eV) photon energy to increase the momentum resolution; also under this photon energy, the bulk states intensity is greatly suppressed and thus the surface states can be easily seen. Indeed, from these fine measurements, the two surface states in the calculations (Fig. 4a–d) are both observed (see Fig. 4e–h, and the

measurements from more photon energies can be found in the Supplementary Fig. 5). Interestingly, as the positions of P1 and P2 vary with binding energy (see Fig. 4a–d), the surface Fermi arcs observed in our measurements (Fig. 4e–h) also vary accordingly (in shape and length), in good agreement with the calculation, thus strongly supporting their topological origin. To further investigate their spin texture, we have also carried out preliminary spin-resolved ARPES measurements that can be found in the Supplementary Note 3 and Supplementary Fig. 6.

## Discussion

Our systematic study on the electronic band structures and the observation of surface Fermi arc states, together with the broad

agreement with the *ab initio* calculations, establish that $T_d$-MoTe$_2$ is a type-II TWS that can not only help understand the puzzling physical properties in the orthorhombic phase of TMDs, but also provide a new platform for the realization of exotic physical phenomena and possible future applications. We note that while we were finalizing this manuscript, three other groups[31–33] also independently studied this family of compounds and the surface states and arcs.

## Methods

**Sample synthesis.** The chemical vapour transport method was employed to grow MoTe$_2$ crystals using polycrystalline MoTe$_2$ powder and TeCl$_4$ as a transport additive. Here, 1 g of polycrystalline powder and TeCl$_4$ (3 mg ml$^{-1}$) were sealed in a quartz ampoule that was then flushed with Ar, evacuated, sealed and heated in a two-zone furnace. Crystallization was conducted from (T2) 1,000 to (T1) 900 °C. The quartz ampoule was then quenched in ice water to yield the high-temperature monoclinic phase, the 1T'-MoTe$_2$. When cooling down below ∼250 K, a structural-phase transition occurs from the centrosymmetric 1T' phase to the noncentrosymmetric T$_d$ phase that is the TWS phase, which was known in the resistivity measurement[25,34].

**Angle-resolved photoemission spectroscopy.** Regular ARPES measurements were performed at the beamline I05 of the Diamond Light Source and BL 10.0.1 of the Advanced Light Source, both equipped with Scienta R4000 analyzers. The spin-resolved ARPES experiments were conducted at the beamline APE of the Elettra synchrotron, equipped with a Scienta DA30 analyzer combined with a VLEED spin detector. The measurement sample temperature and pressure were 10 K and $<1.5 \times 10^{-10}$ Torr, respectively. The angle resolution was 0.2° and the overall energy resolutions were better than 15 meV. The samples were cleaved *in situ* along the (001) plane.

**Ab initio calculations.** The density functional theory calculations have been performed using the Vienna *ab initio* simulation package with the projected augmented wave method[35]. The exchange and correlation energy was considered in the generalized gradient approximation with Perdew–Burke–Ernzerhof (PBE)-based density functional[36]. The energy cutoff was set to 300 eV. The tight binding matrix elements were calculated by projecting the Bloch wave functions to maximally localized Wannier functions[37]. The surface states were calculated from the half-infinite model by using the interative Green's function method[38]. The $k$-dependent local density of states are projected from the half-infinite bulk to the outermost surface unit cell. Experimental lattice constants were used in all the calculations[25].

**Data availability.** The data that support these findings are available from the corresponding author on request.

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

## Acknowledgements

This work is supported by grant from CAS-Shanghai Science Research Center, Grant No: CAS-SSRC-YH-2015-01. This work has been partly performed in the framework of the nanoscience foundry and fine analysis (NFFA-MIUR Italy) project. Y.L.C. acknowledges the support of the EPSRC Platform Grant (Grant No EP/M020517/1) and Hefei Science Center CAS (2015HSC-UE013). C.F. acknowledges the financial support by the ERC Advanced Grant (No 291472 'Idea Heusler'). Advanced Light Source is operated by Office of Basic Energy Science of US DOE (contract DE-AC02-05CH11231). C.-C.H. and J.J. acknowledge the support of the NRF, Korea through the SRC Center for Topological Matter (No 2011-0030787).

## Author contributions

Y.L.C. conceived the experiments. J.J. and Z.K.L. carried out ARPES measurements with the assistance of H.F.Y. and C.C. Z.K.L. and H.F.Y. carried out Spin ARPES measurement with the help of I.V. and J.F. C.R.R. and Y.P.Q. synthesized and characterized the bulk single crystals. B.H.Y. and Y.S. performed *ab initio* calculations. All authors contributed to the scientific planning and discussions.

## Additional information

**Competing financial interests:** The authors declare no competing financial interests.

