## [Peer Review File · Nature Communications]

Reviewers' comments:

Reviewer #1 (Remarks to the Author):

As far as I have read (the revised manuscript seems to be not much improved though), I recommend it to be published in Nature Communications after the following concerns could be removed.

p.2: "However, the complex 3D crystal structure and the large number (12 pairs) of WPs in these first generation TWSs may pose difficulties in the exploration of novel physical phenomena and practical applications."

=> This statement is questionable. The type-II WS has a finite density of states which would be more complicated than that of the type-I. It may be necessary to clarify how the type-II WS would have better aspect.

p.6: "Such observation of the non-degenerate spin structure is consistent with the non-trivial topological nature of the surface Fermi arcs, which will help understanding spin-related physical phenomena in MoTe₂."

=>The cut show in Fig.4c apparently includes both surface states SS and SA, while the authors quote the arc (SA) only. I have to note again that not only surface Fermi arcs but the trivial surface state (SS) should show the spin polarization as already described in Ref [A. Tamai et al., Phys. Rev. X 6, 031021 (2016).]. Also, I have a serious concern that the spin polarization would vary with different photon energy as well as different light polarizations. It means that the spin polarization measurements would not help so much to identify the topology characters in the states unfortunately. I would again like to suggest the authors to remove this part from the manuscript.

Response to Reviewer's comments:

Reviewer #1:

As far as I have read (the revised manuscript seems to be not much improved though), I recommend it to be published in Nature Communications after the following concerns could be removed.

Authors' response:

We thank the reviewer for the recommendation to publication and the helpful suggestions, which we have addressed in the revised manuscript with the explanations below:

p.2: "However, the complex 3D crystal structure and the larger number (12 pairs) of WPs in these first generation TWSs may pose difficulties in the exploration of novel physical phenomena and practical applications."

This statement is questionable. The type-II WS has a finite density of states which would be more complicated than that of the type-I. It may be necessary to clarify how the type-II WS would have better aspect.

Authors' response:

We thank for the Reviewer's comment. Our intention was to emphasize that the 3D crystal structure and the many WPs make the first generation of type-I TWS (TaAs family) less ideal for realizing the unusual physical properties of the TWS. In fact, in the TaAs family of TWSs, there are also normal bulk states overlap (in energy) with the Weyl points, which also introduce further complexity. Thus, in the following two aspects, the MoTe₂ family TWSs show advantages over the TaAs family TWSs:

- (i) The MoTe₂ family TWSs are 2D layered materials, meaning that they're much easier to process (e.g. exfoliation for transport measurements, etc.); and the fact that these family of materials can be grown into high quality thin films by MBE further makes them ideal candidates for the study of unusual physical phenomena of TWS with precise controls (e.g. thickness dependence, strain dependence, etc.). Moreover, these advantages make the MoTe₂ family TWSs more friendly for practical applications (e.g. fabrication of high quality devices, etc.).
- (ii) In MoTe₂ family of TWSs, there are only 4 pairs of WPs, which shows a significant reduction comparing to the 12 pairs of WPs in the TaAs family of TWSs.

On the other hand, it is true that as the reviewer suggested, in type-II TWSs, the tilted Weyl fermions introduce finite density of state at the Fermi-surface, but this also make them different from the type-I TWSs, and can show more (and different) interesting phenomena, such as anisotropic chiral anomaly, etc. Also, recent transport measurements have revealed many interesting properties in MoTe₂ family compounds, including the record high never-saturating magnetoresistance and superconductivity, thus the study of

these properties and their interplay with the topological electronic structure will be of great interest and importance.

Based on the above consideration and taking the reviewer's suggestion, we have revised the manuscript by changing the last sentence of Paragraph 2 into: "However, the complex 3D crystal structure in these first generation TWSs may pose difficulties in the exploration of their practical applications; and their large number (12 pairs) of WPs could make the exploration of novel physical phenomena complicated. Therefore, new TWS materials easier to process, more friendly to device fabrication applications and with less WPs are desired." to avoid confusion.

p.6: "Such observation of the non-degenerate spin structure is consistent with the non-trivial topological nature of the surface Fermi arcs, which will help understanding spin-related physical in MoTe₂."

The cut show in Fig.4c apparently includes both surface states SS and SA, while the authors quote the arc (SA) only. I have to note again that not only surface Fermi arcs but the trivial surface state (SS) would show the spin polarization as already described in Ref [A. Tamai et al., Phys. Rev. X 6, 031021 (2016)]. Also, I have a serious concern that the spin polarizations. It means that the spin polarization measurements would not help so much to identify the topology characters in the states unfortunately. I would again like to suggest the authors to remove this part from the manuscript.

Authors' response:

We thank the Reviewer for the considerate suggestion. Although we have tried our best to choose the photon energy to suppress the bulk band and use the best available instrument resolution to discriminate the SS and SA. Indeed we may not completely exclude the contribution from the SS states, due to the relatively low angle and energy resolution (compared to regular ARPES) of the Spin-measurement (caused by the low spin-detection efficiency).

We thus take the Reviewer's suggestion by removing this part from the main text for clarity. The related discussion on spin measurement has now been moved to the supplementary information, as reference for the more experienced readers.

Summary of revisions

1. Main Text:

- (i) Line 31 (page 1, paragraph 1): Replace "...in agreement with our *ab-initio* calculations." with "...in agreement with our *ab-initio* calculations witch have non-trivial nature."
- (ii) Line 32 (page 1, paragraph 1): Delete "From spin-resolved ARPES measurements, we demonstrate the non-degenerate spin-texture of surface Fermi arcs, thereby proving their non-trivial topological nature."
- (iii) Line 55 (page 2, paragraph 2): Replace "However, the complex 3D crystal structure and the large number (12 pairs) of WPs in these first generation TWSs may pose difficulties in the exploration of novel physical phenomena and practical applications." with "However, the complex 3D crystal structure in these first generation TWSs may pose difficulties in the exploration of their practical applications; and their large number (12 pairs) of WPs could make the exploration of novel physical phenomena complicated. Therefore, new TWS materials easier to process, more friendly to device fabrication applications and with less WPs are desired."
- (iv) Line 71 (page 3, paragraph 2): Add ", which are topological non-trivial" before "...and our ab-initio calculation."
- (v) Line 72 (page 3, paragraph 2): Delete "Moreover, by spin-resolved ARPES measurement, we observed the non-degenerate spin texture of the surface Fermi arcs, further supporting their non-trivial topological nature."
- (vi) Line 140 (page 5, paragraph 2): Add ", thus strongly support that their topological origin. To further investigate their spin texture, we have also carried out preliminary spin-resolved ARPES measurements, which can be found in the Supplementary information." before "in agreement with the calculation".
- (vii) Line 142 (page 5, paragraph 3): Delete the whole paragraph from line 142 to line 154.
- (viii) Line 157 (page 5, paragraph 4): Delete "and its spin texture,".

2. Figures and captions

- (i) Delete Fig. 4c and Fig. 4d, and its corresponding captions.

3. References

- (i) Delete references 31-33.

4. Supplementary Information:

- (i) Add part E: spin polarization of the surface arc states together with Fig. S6.